# A Comparison of the Effect of a 4.4-MHz Radiofrequency Deep Heating Therapy and Ultrasound on Low Back Pain: A Randomized, Double-Blind, Multicenter Trial

**DOI:** 10.3390/jcm11175011

**Published:** 2022-08-26

**Authors:** Jung Hwan Lee, Jong Geol Do, Hee Jin Park, Yong-Taek Lee, Sang Jun Kim

**Affiliations:** 1Namdarun Rehabilitation Clinic, Yongin 16858, Korea; 2Department of Physical and Rehabilitation Medicine, Samsung Medical Center, Sungkyunkwan University School of Medicine, Seoul 06351, Korea; 3Department of Radiology, Kangbuk Samsung Hospital, Sungkyunkwan University School of Medicine, Seoul 03181, Korea; 4Department of Physical and Rehabilitation Medicine, Kangbuk Samsung Hospital, Sungkyunkwan University School of Medicine, Seoul 03181, Korea; 5Seoul Jun Rehabilitation Clinic, Seoul Jun Rehabilitation Research Center, Seoul 06737, Korea

**Keywords:** radiofrequency, diathermy, ultrasound, physical therapy, low back pain

## Abstract

Despite the increasing interest in RF (Radiofrequency) therapy, little is known about its effectiveness for low back pain (LBP). The aim of this study was to investigate the effectiveness of 4.4-MHz RF diathermy compared to ultrasound (US) in patients with LBP. One-hundred-and-eighteen patients with LBP were randomized with RF (*n =* 62) or US (*n =* 56). Investigator and subjects were blinded to the treatment group. Either RF (4.4 MHz, 45 W/cm^2^) or US (1 MHz, 2 W/cm^2^) was applied for 10 to 15 min, 3 times per week for 4 weeks. The primary outcome was the Oswestry Disability Index (ODI, %). Secondary outcomes were numeric rating scale (NRS), Biering–Sorensen test, up-and-go test, successful pain relief, and successful functional improvement. Clinical outcomes were evaluated prior to intervention (baseline), and at 4 and 12 weeks after treatment. There were no significant differences between the groups regarding baseline demographic and clinical characteristics. Both groups observed a significant improvement of ODI (%), NRS, Biering–Sorensen test, and up-and-go test at 4 and 12 weeks after treatment (*p* < 0.05); however, no significant differences were found between groups. The RF group showed a higher proportion of successful pain relief at 12 weeks after treatment than the US group (*p* = 0.048). The RF diathermy showed favorable results in pain reduction, improvement of function, mobility, and back muscle endurance. Compared with US, RF diathermy obtained slightly better perception of patients in pain relief at 12 weeks after treatment. The results from this study indicated that 4.4-MHz RF diathermy can effectively be used as a conservative treatment option for patients with LBP.

## 1. Introduction

Low back pain (LBP, ICD-10: M54.5) is a major cause of morbidity and affects 80–85% of people over their lifetime [1]. LBP is a common health and socioeconomic problem and a major public health burden [2]. The treatment goal for LBP is to relieve pain, improve function, and promote a return to the patient’s desired level of daily activity [3]. Various treatments include pharmacologic, exercise therapy, psychological therapy, and physical therapeutic modality including ultrasound, transcutaneous electrical stimulation, low-level laser therapy, and superficial or deep heat therapy are available [4]. Previous practice guidelines recommended biopsychosocial management with initial non-pharmacological treatment and less emphasis on medication use and surgery [5]. Local heat therapy is a commonly used physical therapeutic modality in patients with LBP [6]. Despite its common use, there is still inconclusive evidence regarding which type of heat therapy is superior for LBP.

Therapeutic ultrasound (US) is a commonly used noninvasive treatment tool for LBP. US with a frequency of 1–3 MHz is a frequently used deep heat therapy for LBP. Therapeutic US has thermal and mechanical effects on deep tissues via the delivery of ultrasonic energy [7]. The increased molecular motion created by acoustic waves increases tissue temperature, which leads to pain reduction through alterations in conduction velocity, reduced muscle spasm, and increased collagen tissue extensibility and local blood flow [8]. In addition, ultrasonic pulsating waves cause acoustic cavitation and microbubbles, which are non-thermal effects that also contribute to pain reduction by facilitating cell membrane activity, vascular wall permeability, and soft tissue healing [6,9]. For these reasons, therapeutic US was found to be effective for pain reduction, functional improvement, and improved lumbar range of motion in patients with LBP [9,10].

Transcutaneous radiofrequency (RF) diathermy is a noninvasive modality that consists of the emission of high-frequency electromagnetic waves [11]. RF diathermy is used to reduce pain and inflammation and enhance tissue healing. The electromagnetic energy delivered by RF diathermy causes a rise in tissue temperature, which can lead to pain relief and decreased inflammation by enhancing blood flow or oxygen uptake and accelerating cellular activities and metabolism [11,12]. In this regard, RF diathermy is used to manage pain and reduce the recovery time in musculoskeletal disorders [13]. Wavelengths of 1–100 MHz have also been applied to treat LBP patients [11,14]. However, the clinical benefit of RF diathermy in treating LBP patients has not been established. A previous study reported that RF diathermy was inferior to exercise programs in terms of pain control and functional tests. However, since the study participants were hockey players or young men, the results of this study cannot be generalized to other populations [15,16]. In clinical practice, RF diathermy is expected to become a popular treatment method, because it has the advantages of low cost, little patient discomfort, and minimal side effects. In the study of chronic LBP, RF diathermy showed a higher efficacy with respect to superficial heat therapy in reducing Oswestry Disability Index (ODI) [17]. Previous randomized controlled study revealed that continuous RF diathermy was found to be more effective in reducing pain in patients with chronic low back pain than placebo or pulsed RF diathermy [18]. Since the degree of penetration into the tissue varies depending on the RF wave frequency, it is important to set the optimal RF wave frequency. RF waves for pain management in the clinical setting average 448 kHz, while microwaves usually range from 915 MHz to 2450 MHz [11,19,20]. However, it is not yet clear which wave frequency is effective for pain management. Previous studies reported that the therapeutic effect of a 4.4-MHz RF on muscle injury in a rat model. A 4.4-MHz RF elevated the muscle layer temperature without inducing cellular damage [21]. Furthermore, a 4.4-MHz RF showed therapeutic effect on muscle contusion by reducing muscle swelling in a rat model [22]. This has the advantage of a tissue penetration without tissue damage and an anti-inflammation action in the target tissue [21,22]. We hypothesized that a 4.4-MHz RF diathermy would be effective to decrease pain and improve physical function in patients with LBP. 

The effectiveness of RF diathermy on the function and pain in the LBP is controversial [15,16,17,18]. To the best of our knowledge there is no prospective, large sample size (over than 100 participants) study investigating the effectiveness of RF diathermy in the treatment LBP. The aim of this study was to investigate the clinical effectiveness of 4.4-MHz RF diathermy on disability, pain, back muscle endurance, mobility, and satisfaction, and to compare it with the US treatment in patients with LBP.

## 2. Methods 

### 2.1. Study Design and Participants

A double-blind, randomized prospective trial was conducted with 118 patients in three hospitals of a musculoskeletal clinic. This study was approved by the Institutional Review Board of the authors’ affiliated institutions and the Ministry of Food and Drug Safety (SMC2015-12-109-003). Eligible patients included those from 19 years to 70 years old who visited the Department of Physical and Rehabilitation Medicine between March 2017 and October 2017 with a diagnosis of LBP (pain duration ranged from 1 to 6 months). Among them, we selected patients who provided written informed consent and satisfied the following criteria: (1) pain location ranging from T12 to the buttocks, (2) no radicular pain or signs of radiculopathy on physical examination of the lower extremity within 3 days, (3) no previous lumbar surgery, (4) no history of spinal injection within 2 months, (5) ODI(%) of ≥30%, (6) numeric rating scale (NRS) ≥ 4, (7) no evidence of recent bone fracture or tumor in simple lumbar radiography, and (8) no skin diseases, including contact dermatitis, that could produce skin problems related to application of the physical therapy apparatus or gel. Exclusion criteria included: (1) neurologic deficits of the lower extremities, (2) women who were pregnant, (3) high fever (≥38 °C), (4) abnormalities in cell blood count (platelet count < 50,000/mm^3^), (5) cardiac problems or pacemaker implant, (6) metal implant within bony structures, and (7) severe cognitive or communication impairment that would prevent appropriate responses to the questionnaire. This study was registered under identifier KCT0002734 (https://cris.nih.go.kr, accessed on 1 February 2017). 

### 2.2. Random Assignment and Blinding Technique

If a patient met all eligibility criteria, patients were randomly assigned to either the RF diathermy group or the US group using a computer-generated randomization technique. One experienced clinical research operator randomly allocated the patients using a random 6 blocks (for example, AAABBB, AABABB, etc.) with age stratification (19 to 39 years old, 40 to 59 years old, and 60 to 70 years old). The permuted block randomization according to these age group minimizes the potential for unequal distribution of age between groups. The randomization was conducted without consideration of hospital allocation. The allocation information of the patient was delivered to physical therapists in each hospital without notifying the assessors or the patients, according to the double-blind method. The researcher responsible for randomization was independent from the assessors, assuring blindness to treatment allocation and randomization procedures. For the blinding technique, to prevent the patients from knowing which machine (RF or US) was being used, both machines were covered with clothes, and one idle RF plate was additionally attached to the US machine. During the US treatment, the idle RF plate and a real US probe were simultaneously applied to the patients in the US group. The group randomization sequence was directly communicated to the physical therapist who performed the treatment, while both the assessors and the patients were blinded to assignment. The blinded assessor performed the baseline and post-treatment clinical evaluations. 

### 2.3. Intervention

Treatment of both groups was conducted 3 times per week for 4 weeks. In both groups, the duration of each treatment was 10 to 15 min. The RF group was treated using a HIPER-500 diathermy apparatus^®^ (JS-ON corporation, Seoul, South Korea) (Appendix A). The peak power of this apparatus was 45 W/cm^2^ (±20%) using 500 Ω, and it was operated at 4.4 MHz. The rationale for the use of this apparatus has been reported for muscle injury and shoulder pain [21,23]. This apparatus consisted of a body and two electrodes, a negative electrode with a diameter of 100 mm and a positive electrode with a diameter of 70 mm. Each electrode was made of aluminum and had a polyamide coating that acted as a dielectric medium, insulating its metallic body from the skin surface, thus forming a capacitor with the treated tissues. A manufacturer-supplied conductive cream was employed as a coupling medium between the electrode and the skin surface. The US group was treated with an ultrasound apparatus, Ultrasonic SUS-2N^®^ (SHIN JIN, Seoul, South Korea). The frequency was set to 1 MHz and the power output to 2 W/cm^2^. One idle RF electrode was attached to the RF machine, and the exterior parts of the RF and US apparatus were covered to prevent identification by the participants. Treatment implementation time in this study was the same in both groups. The 4.4-MHz RF diathermy in this study is approximately 10 times as expensive as the US device.

In this study, RF or US treatment was applied to each group alone, and other treatments including exercise therapy were not administered together.

### 2.4. Outcome Measures

Patients were assessed at three measurement sessions: before the start of treatment (baseline evaluation), 4 weeks after treatment, and 12 weeks after treatment. The primary outcome measure was the Korean version of the ODI (%) [24]. The ODI is one of the most commonly used outcome measures for LBP. The ODI is a self-administered questionnaire which consists of 10 items that evaluate the level of pain and interference with several physical activities [25]. The secondary outcomes were NRS, Biering–Sorensen test, up-and-go test, successful pain relief, and successful functional improvement. The pain intensity was evaluated by means of NRS ranging from 0 (no pain) to 10 (worst imaginable pain). A Biering–Sorensen test was performed to evaluate isometric endurance and strength of the back extensor muscles. The patient was positioned prone on a treatment couch with the upper edge of the iliac crests aligned with the edge of the couch. The lower body was fixed to the couch by two straps, located at the level of the greater trochanter of the femur and at the ankles as close to the malleoli as possible. While the participants were secured into position, they were allowed to rest their upper body on a stool for comfort and to minimize fatigue. At the start of the test, the participants placed their arms diagonally across their chest and maintained a neutral position without any support to the upper body for as long as possible. The duration (s) for which the position could be held was measured using a stopwatch. Termination of the test occurred as follows: excessive fatigue, downward sloping of the trunk by more than 10° as observed by visual inspection, unendurable pain, or when four minutes was reached. If the participant’s horizontal position dropped, they were asked to regain horizontal alignment until it could no longer be successfully performed [26,27]. The up-and-go test was performed to assess gait speed and balance performance. Patients were asked to stand up from the chair when the examiner said ‘go’, walk a 3-m distance at normal pace, and then turn back to the chair. The time (s) required to complete this test was measured [28]. Successful pain relief was defined as 50% or greater reduction in the NRS compared with pretreatment. Successful functional improvement was defined as at least 40% reduction in ODI [29]. These evaluations were performed by an assessor who was blinded to individual assignment at pre-treatment, 4 weeks after treatment, and 12 weeks after treatment. 

In addition, the North American Spine Society (NASS) 4-point patient satisfaction index was obtained at 4 weeks after treatment and 12 weeks after treatment to evaluate the degree of patients’ subjective satisfaction. The choices provided include: (1) “The treatment met my expectations”; (2) “I did not improve as much as I had hoped, but I would undergo the same treatment for the same outcome”; (3) “I did not improve as much as I had hoped, and I would not undergo the same treatment for the same outcome”; and (4) “I am the same or worse than before treatment” [30]. The number of patients who needed to take pain medication during the follow-up period was evaluated. Medications that were taken for pain control before participation in this study were assessed, and the patients were asked to cease taking these medications before participating in this study. However, the same kind of non-steroidal anti-inflammatory drugs (aceclofenac 100 mg twice a day) was prescribed and permitted to be taken during treatment and follow-up periods when back pain was severe enough to require medication. These steps allowed control of medications taken by all patients, as well as enabling assessment and comparison of required medications, thereby indirectly measuring the degree of clinical improvement between the two groups.

### 2.5. Sample Size Calculation

Sample size was calculated based on the literature comparing high-intensity laser and ultrasound in LBP [31]. A difference of at least 2.73 in the ODI was considered statistically meaningful. Standard deviation was established as 4.44, based on the standard deviation observed between the two groups (1.9 and 4.0). Considering a 0.05 two-sided significance level, a power of 80%, and an allocation ratio of 1:1, at least 42 patients in each group were required. However, considering the approximate dropout rate was estimated as 30%, 60 patients in each group was the recruitment goal.
$$N=2\times \frac{\sigma^{2}\times {\left(Z_{\alpha }+Z_{\beta }\right)}^{2}}{d^{2}}=2\times 4.44^{2}\times {\left(1.96+0.84\right)}^{2}/2.73^{2}=41.53$$

### 2.6. Statistical Analysis

The SPSS version 14.0 statistical package (SPSS Inc., Chicago, IL, USA) was used for statistical analysis. Continuous variables are presented as the mean and standard deviation (SD). Frequency count and percentage are presented for categorical variables. Chi-square with Fisher’s exact test was used to compare age distribution, gender, occupation, successful rate of NRS and ODI (%), and the NASS patient satisfaction index after treatment between the two groups. Independent *t*-tests were performed to identify differences in age, NRS, ODI (%), number of treatment sessions performed, and number of patients who required pain medications during the treatment and follow-up periods. Paired *t*-tests were performed to assess clinical improvement regarding NRS, ODI (%), Biering–Sorensen test, and the up-and-go test in each group. Repeated measures analysis of variance (RM-ANOVA) was used to compare improvements in NRS, ODI (%), Biering–Sorensen test, and the up-and-go test between the RF and US groups. Results were considered statistically significant at a *p* value less than 0.05.

## 3. Results

### 3.1. Baseline Characteristics

One-hundred-and-twenty-six patients were recruited, but eight patients were excluded during screening: 4 patients refused to provide informed consent, and 4 patients were not eligible for the inclusion criteria. A total of 118 patients who provided written informed consent were randomly allocated into two groups, an RF group and a US group. Sixty-two patients (49 females, 79.0%) in the RF group and 56 patients (43 females, 76.8%) in the US group were included in this study. Fourteen patients were dropped because the treatment was not completed. Completion of treatment was regarded as participating in 8 of the 12 treatment sessions offered. Figure 1 illustrates the study flow diagram. 

The mean age of the patients in the study was 47.7 ± 13.3 years. There were no significant differences between groups regarding age, gender, weight, height, and proportions of occupation (Table 1). Baseline clinical parameters were not found statistically significantly different between groups in terms of ODI (%), NRS, Biering–Sorensen test, and the up-and-go test (Table 2). The number of treatment sessions performed for the RF group and US group was 10.9 ± 2.42 and 11.02 ± 1.59, respectively, which represented a non-significant difference.

### 3.2. Clinical Evaluation

The changes in clinical and functional parameters in both groups are shown in Table 3. Initial ODI (%) was 46.06 ± 13.94 in the RF group and 44.33 ± 14.68 in the US group, which was not significantly different. After RF treatment, ODI (%) decreased to 20.61 ± 11.75 at 4 weeks and 19.00 ± 12.06 at 12 weeks, while ODI decreased to 22.42 ± 13.56 at 4 weeks and 20.44 ± 15.21 at 12 weeks after US treatment. Both the RF and US groups exhibited a significant reduction of ODI (%), but no significant difference of ODI was observed between the two groups. After stratifying ODI results by age, the change of ODI (%) in the RF group was not significantly different from that of ODI in the US group in all age groups.

After RF treatment, NRS decreased to 3.11 ± 2.00 at 4 weeks and 2.58 ± 1.96 at 12 weeks, while NRS decreased to 3.25 ± 1.75 at 4 weeks and 2.86 ± 1.76 at 12 weeks after US treatment. Both the RF and US groups exhibited a significant reduction of NRS, but no significant difference of NRS was observed between the two groups.

The Biering–Sorensen test was improved to 32.02 ± 29.69 s at 4 weeks and 33.20 ± 30.09 s at 12 weeks in the RF group, while it was improved to 28.66 ± 26.23 s at 4 weeks and 29.66 ± 33.37 s at 12 weeks in the US group. The Biering–Sorensen test showed a significant difference according to the visit, but not between groups. The up-and-go test was improved to 7.90 ± 2.18 s at 4 weeks and 8.15 ± 2.40 s at 12 weeks in the RF group, while this was improved to 8.33 ± 2.34 s at 4 weeks and 8.43 ± 2.42 s at 12 weeks in the US group. The up-and-go test showed a significant difference according to the visit, but not between groups.

Table 4 shows the results in terms of patient satisfaction on treatment and pain medication administered during the study. The RF group had a significantly lower proportion of patients who required pain medication during treatment than the US group (*p =* 0.048). Both groups showed good patient satisfaction (NASS 1 or 2); however, no significant difference was observed in NASS between the two groups at 4 and 12 weeks after treatment. During treatment, no serious side effects were observed in either group.

## 4. Discussion

This study evaluated the effectiveness of a 4.4-MHz RF diathermy in comparison to US for patients with LBP. In this study, both the RF and US groups showed significantly improved ODI and NRS, as well as improvements on the Biering–Sorensen test and the up-and-go test at 4 weeks and 12 weeks after treatment. In this study, the 4.4-MHz RF diathermy group revealed a higher rate of successful pain relief at 12 weeks after treatment than the US for LBP, but the RF group did not show a significant difference in functional improvement compared to the US group. The results suggest that the 4.4-MHz RF diathermy has comparable pain and functional improvement effects as US in LBP patients. In terms of perception of pain relief, the 4.4-MHz RF diathermy was more effective than US treatment, the 4.4-MHz RF diathermy can be applied to patients with LBP in clinical practice.

Local heat applications are an effective treatment modality in LBP patients, reducing pain and improving function and range of motion [7,9,10,32]. However, the effectiveness of RF diathermy in the management of LBP has not been established. Previous studies demonstrated better clinical outcomes using US along with exercise than using exercise alone, in addition to demonstrating superiority of the US over placebo. A meta-analysis study concluded that US is beneficial for low back function, but the clinical benefits were short term [33]. RF diathermy is used for LBP to relieve pain, but the clinical effect of RF diathermy on LBP patients has not yet been established. Previous randomized controlled studies revealed that RF diathermy (27.12 MHz, short wave) resulted in significantly less clinical efficacy than stabilization exercises or manual therapy [15,16,34]. Studies comparing RF diathermy with other treatments, such as spinal manipulation, exercise, traction, and sham treatment, have shown that RF diathermy did not yield better clinical outcomes compared to other modalities [35,36]. In a previous study, 49 patients of LBP were treated 450 KHz RF diathermy or superficial heating therapy. Both groups showed effective in pain reduction, 450 KHz RF diathermy allowed better result than superficial heating therapy in improving ODI [17]. In our study, there was reduced pain and improved function in both groups. There was no significant difference between the groups in terms of ODI (%), NRS, but there was significant successful pain relief in the 4.4-MHz RF diathermy group. We considered that previous studies used relatively high frequency diathermy, the physical effects of which were limited to superficial tissues and did not penetrate deep tissues such as back muscle. RF diathermy with a frequency range of 4.4 MHz is a relatively low frequency modality and can therefore penetrate deep tissues. In a rat model, RF diathermy with a frequency range of 4.4 MHz demonstrated elevated tissue temperature from the skin to the muscle layer without both histologic change and apoptosis [21]. For the treatment of LBP, this wave frequency might be the optimal setting for RF diathermy to penetrate deep tissue without muscle injury.

In the present study,4.4-MHz RF diathermy was used using CET (capacitive electrical transfer) and RET (resistive electrical transfer). CET was a method that allowed RF energy to be transferred by the principle of creating an electric capacitor involving the patient’s body via a moveable external application electrode. The CET method had limitations in that it generated a temperature increase mainly on the superficial area near the electrodes and poorly transferred heat energy to deeper areas [37]. The other method, RET, varied from CET in that the application electrode was not insulated, allowing the current to be transferred directly to the patient. As a result, the energy was less dissipated and penetrated more deeply so that it could easily generate a temperature increase in deep tissues [38]. The property of this RF diathermy device using CET and RET simultaneously with 4.4 MHz overcame the previous limitations of RF diathermy, which is expected to be more useful for patients with LBP by inducing effective physiological changes. One study regarding the histological and temperature change by 4.4-MHz RF diathermy in Sprague-Dawley rats demonstrated that a significant rise in temperature was found at a 1-cm depth as well as at the skin surface, and no histological damage or degeneration was observed [21]. A study of human participants also reported that RF diathermy using CET and RET raised tissue temperature and increased total and oxygenated hemoglobin in deep and superficial locations. This suggested that CET and RET succeeded in delivering heat energy and increasing blood circulation and tissue metabolism in both deep and superficial tissues [38]. The 4.4-MHz RF diathermy showed improvement of physical performance and relief of pain in shoulder patients [23]. In this regard, 4.4-MHz RF diathermy using CET and RET was presumed to produce more clinical efficacy than US without any harmful effects. In line with this, our study supports the clinical efficacy of 4.4-MHz RF diathermy, which yields a higher successful pain relief rate after 12 weeks of treatment compared to US.

It was notable that RF diathermy as well as US led to increased back muscle endurance, demonstrated by improvement in the Biering–Sorensen test. Decreased strength or endurance in LBP patients is explained by the fear avoidance belief, which means that patients with pain try to avoid movement due to the fear of pain [39]. Decreased muscle endurance was also a result of accumulation of metabolite waste and the inability to provide adequate blood circulation for oxygen supply to the tissue [40]. In addition, muscle spasm in patients with LBP was another cause of deceased muscular endurance [9]. RF diathermy and US can improve blood circulation; therefore, inflammation or muscle spasms are reduced, and, consequently, pain is reduced, contributing to increased muscle endurance.

There are several limitations in this study. First, the patient should not know which modality (RF or US) was applied, but the two modalities do not perform in a similar way. To maintain blindness, we covered the machines with clothes, and during treatment, the idle plate and a real probe were simultaneously applied to the patients. Second, we enrolled patients with LBP between 1 month to 6 months, except for the acute LBP. As patients subacute and chronic LBP were included, the study population of this study is not homogenous. Third, the physical and physiology features that causes LBP may differ between men and women. In this study were predominantly women in both groups, from this it is difficult to generalize. A limitation in this study is that it was predominantly female in both groups. Lastly, it is a limitation of the study that this study did not prohibit of oral pain medication. However, we showed administered pain medication as a secondary outcome. Furthermore, as many LBPs are chronic, further studies should investigate the effectiveness and safety for long-term use.

The strength of this study was that the relatively large sample size, double blind, randomized study design provided a better understanding of the clinical efficacy of RF diathermy in LBP. In our study, RF diathermy and US were applied as monotherapy rather than in combination with other treatments (i.e., exercise and/or manual therapy). Through this study design, we tried to avoid the conflicting effects of different treatments in order to weigh the effectiveness of each single treatment. Considering the results and strengths of this study, a 4.4-MHz RF diathermy seems to be effective for the treatment of LBP, and it can be considered as a conservative treatment for LBP.

## 5. Conclusions

The 4.4-MHz RF diathermy showed favorable results in pain reduction, as well as improvement of function, mobility, and back muscle endurance, which was similar to US treatment. The 4.4-MHz RF diathermy had a slightly better perception of patients in pain relief at 12 weeks after treatment. The clinical findings suggest that 4.4-MHz RF diathermy may be used as a conservative treatment option for patients with LBP.

## Figures and Tables

**Figure 1 jcm-11-05011-f001:**
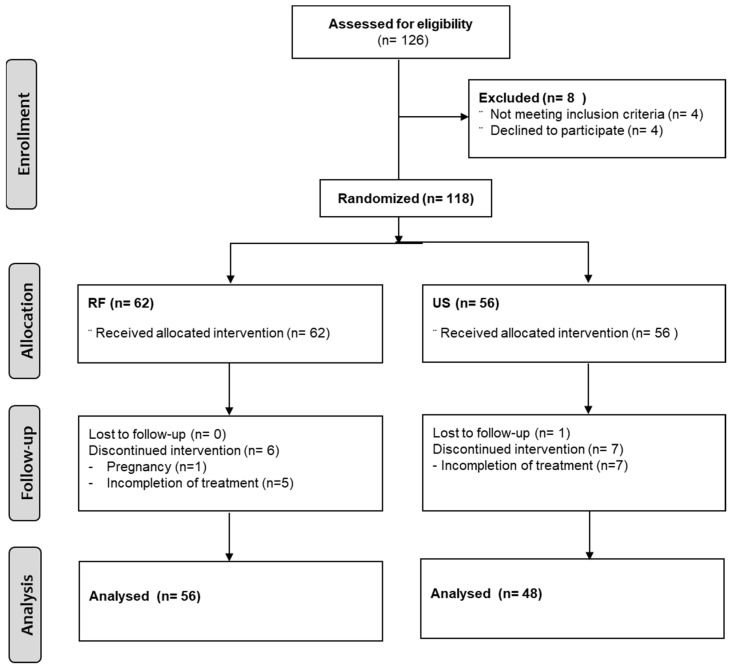
The flow diagram of the study. A total of 118 patients were randomly allocated into the radiofrequency (RF) or the ultrasound (US) group, with 62 patients in the RF group and 56 patients in the US group.

**Figure 2 jcm-11-05011-f002:**
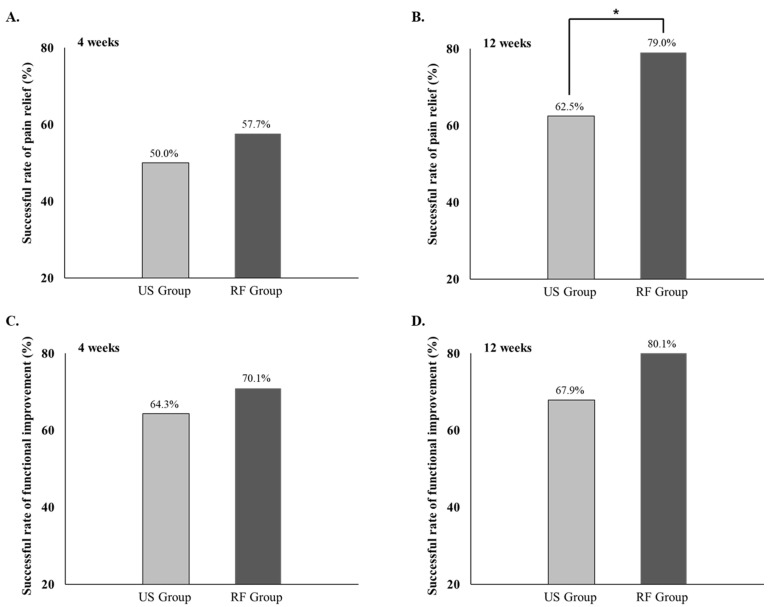
Successful rate of pain relief and functional improvement in both groups. The radiofrequency (RF) group showed a significantly higher rate of successful pain relief at 12 weeks after treatment compared to the ultrasound (US) group (*p* < 0.05), while no significant difference was found at 4 weeks after treatment. (**A**) Successful rate of pain relief at 4 weeks, (**B**) successful rate of pain relief at 12 weeks, (**C**) successful rate of functional improvement at 4 weeks, (**D**) successful rate of functional improvement at 4 weeks.* *p* < 0.05.

**Table 1 jcm-11-05011-t001:** Demographic characteristics of the patients.

		RF	US	Total	*p* Value
		*n* = 62 (%)	*n* = 56 (%)	*n* = 118	
Gender	Male	13 (21.0)	13 (23.2)	26 (20.0)	0.7339
	Female	49 (79.0)	43 (76.8)	92 (78.0)	
Age (years)	Mean ± SD	46.5 ± 14.0	48.9 ± 12.5	47.7 ± 13.3	0.3038
	19 to 39	22 (34.9)	15 (23.8)	37 (31.1)	0.7738
	40 to 59	30 (47.6)	31 (49.2)	61 (51.3)	
	60 to 70	11 (17.5)	10 (15.9)	21 (17.7)	
Height (cm)	Mean ± SD	162.9 ± 6.5	162.2 ± 7.7	162.6 ± 7.1	0.2837
Weight (kg)	Mean ± SD	59.7 ± 8.7	60.8 ± 8.2	60.2 ± 8.5	0.5382
Occupation	Employed	20 (31.7)	22 (39.3)	42 (35.3)	0.4104
	Official	18 (28.6)	10 (17.8)	28 (23.5)	
	Homemaker	22 (34.9)	23 (41.1)	45 (37.8)	
	Other	3 (4.8)	1 (1.8)	4 (3.4)	

Data are mean ± SD or *n* (%) values. RF, radiofrequency group; US, ultrasound group; SD, standard deviation. Chi-square with Fisher’s exact test was used to compare gender, age distribution, occupation. Independent *t*-tests were performed to identify differences in age, height, weight.

**Table 2 jcm-11-05011-t002:** Baseline clinical parameters in both groups.

	RF (*n* = 62) (95% CI)	US (*n* = 56) (95% CI)	*p* Value
ODI (%)	46.06 ± 13.94 (42.52–49.60)	44.33 ± 14.68 (40.40–48.26)	0.5123
NRS	6.21 ± 1.33 (5.87–6.55)	5.84 ± 1.29 (5.49–6.18)	0.1285
Biering–Sorensen test (s)	16.74 ± 18.17 (12.13–21.36)	18.53 ± 15.31 (14.39–22.66)	0.5696
Up-and-go test (s)	9.07 ± 2.38 (8.47–9.68)	9.25 ± 2.70 (8.53–9.98)	0.7024

Data are mean ± SD (95% CI) values. RF, radiofrequency group; US, ultrasound group; ODI, Oswestry Disability Index; NRS, numeric rating scale; CI, confidence interval.

**Table 3 jcm-11-05011-t003:** Changes in clinical and functional parameters in both groups.

			RF (*n* = 62)						US (*n* = 56)					*p* Value ^b^	*p* Value ^c^
		Mean	± SD	Min	Max	*95% CI*	*p* Value ^a^	Mean	± SD	Min	Max	*95% CI*	*p* Value ^a^		
ODI (%)	Baseline	46.06	13.94	31.11	75.56	42.52–49.60		44.33	14.68	31.11	75.56	40.40–48.26		0.512	
	At 4 weeks	20.61	11.75	0.00	53.33	17.62–23.59	<0.001	22.42	13.56	0.00	73.33	18.79–26.05	<0.001	0.439	
	At 12 weeks	19.00	12.06	0.00	53.33	15.93–22.06	<0.001	20.44	15.21	0.00	73.33	16.36–24.51	<0.001	0.568	0.367
NRS	Baseline	6.21	1.33	4.00	9.00	5.87–6.55		5.84	1.29	4.00	8.00	5.49–6.18		0.129	
	At 4 weeks	3.11	2.00	0.00	9.00	2.60–3.62	<0.001	3.25	1.75	0.00	7.00	2.78–3.72	<0.001	0.694	
	At 12 weeks	2.58	1.96	0.00	9.00	2.08–3.08	<0.001	2.86	1.76	0.00	7.00	2.39–3.33	<0.001	0.424	0.118
Biering–Sorensen test (s)	Baseline	16.74	18.17	0.00	88.00	12.13–21.36		18.53	15.31	0.00	74.00	14.39–22.66		0.570	
	At 4 weeks	32.02	29.69	0.20	150.89	24.48–39.56	<0.001	28.66	26.23	0.00	134.21	21.63–35.68	0.001	0.518	
	At 12 weeks	33.20	30.09	0.00	179.81	25.56–40.84	<0.001	29.66	33.37	0.00	210.00	20.72–38.60	0.004	0.545	0.521
Up-and-go test (s)	Baseline	9.07	2.38	5.60	17.00	8.47–9.68		9.25	2.70	4.12	18.00	8.53–9.98		0.702	
	4 weeks	7.90	2.18	4.60	15.00	7.35–8.45	<0.001	8.33	2.34	4.80	14.00	7.70–8.96	<0.001	0.303	
	12 weeks	8.15	2.40	5.06	17.00	7.54–8.76	<0.001	8.43	2.42	4.60	16.00	7.78–9.08	<0.001	0.528	0.669

RF, radiofrequency group; US, ultrasound group; SD, standard deviation; CI, confidence interval; ODI, Oswestry Disability Index; NRS, numeric rating scale; ANOVA, analysis of variance. ^a^ Paired *t*-test (within groups, baseline vs. at 4 weeks, baseline vs. at 12 weeks), ^b^ Independent *t*-test (between groups, RF vs. US), ^c^ Time effect-Repeated measures of ANOVA between groups analysis was applied: Baseline, 4 and 12 weeks after treatment between groups. Figure 2 shows the successful rate of pain relief and functional improvement on treatment completion after 4 and 12 weeks. In terms of successful outcomes in NRS, the RF group showed a significantly higher rate of successful pain relief (79.0%) at 12 weeks after treatment compared to the US group (62.5%, *p =* 0.048). There was no significant difference in the proportion of successful functional improvement between the two groups at 4 and 12 weeks after treatment.

**Table 4 jcm-11-05011-t004:** NASS patient-satisfaction index and intake of pain medication.

	RF (*n* = 62)	US (*n* = 56)	*p* Value
	N (%)	N (%)	
NASS at 4 weeks			
1	35 (56.45)	28 (50.00)	
2	20 (32.26)	20 (35.71)	
3	3 (4.84)	4 (7.14)	
4	0 (0.00)	0 (0.00)	0.783
NASS at 12 weeks			
1	37 (59.68)	30 (53.57)	
2	17 (27.42)	16 (28.57)	
3	4 (6.45)	6 (10.71)	
4	0 (0.00)	0 (0.00)	0.658
Pain medication administered			
Yes	4 (6.45)	14 (25.00)	
No	58 (93.55)	42 (75.00)	0.048

Data are *n* (%) values. RF, radiofrequency group; US, ultrasound group; NASS, North America Spine Society 4-point patient satisfaction index.

## Data Availability

The datasets generated and analyzed during this study are not publicly available due to privacy protection and medical confidentiality but are available from the corresponding author for reasonable requests and with patient permission.

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
