# Peer review of "A Comparison of the Effect of a 4.4-MHz Radiofrequency Deep Heating Therapy and Ultrasound on Low Back Pain: A Randomized, Double-Blind, Multicenter Trial"

_jcm, 2022, doi:10.3390/jcm11175011_

Round 1

Reviewer 1 Report

Title: Effect of a 4.4-MHz Radiofrequency Deep Heating Therapy on Low Back 2 Pain: A Randomized, Double-Blind, Multicenter Trial

Although this manuscript was written well and has some interesting ideas, there are several significant flaws that need to be edited.

The main problem of the manuscript was the study design. The authors stated out the manuscript was designed as a double-blind, randomized controlled trial aimed to investigate the clinical effectiveness and safety of 4.4-MHz 100 RF diathermy in the treatment of patients with LBP and compare this with the clinical effectiveness of US treatment. However, no control or placebo group was involved. Moreover, I think the ultrasonography group was not a control group, it was an active intervention group. So, the study mainly did not focus on the effect of a 4.4-MHz Radiofrequency Deep Heating Therapy on low back pain. I think the study compares the effect of 4.4-MHz Radiofrequency Deep Heating Therapy and ultrasonography.  According to the study design, the authors can only prove the comparison of Radiofrequency Deep Heating Therapy and ultrasonography. Therefore, the study design should be edited for all sections of the manuscript.

Title

-The title should be revised as ‘’A comparison of the effects of ………………………’’, if a control or placebo group isn’t involved.

Abstract

- RF was written directly as an abbreviation.

-P values should be added.

Introduction

- The introduction was well explained. The authors may mention more of the studies investigating the effectiveness of RF in low back pain. The difference and importance of the current study from similar studies can be explained more clearly.

-The purpose of the study should be revised.

Methods

-I think, the study did not involve any control group, so line 106 should be corrected.

-Why ages of patients limited from 19 to 70 years? Did the authors have any special causes?

-Was there a special reason for the ODI score >30% in the inclusion criteria of the patients? Please explain.

-Authors stated that patients with 1 to 6 months of low back pain were included in the study. Why did they choose 1 to 6 months? Studies generally include chronic low back pain (more than 3 months) or acute/subacute (less than 3 months). The study did not have a homogenous LBP population.

-Were the physical activity level or exercise habits of participants available? Were the groups equal physical levels?

-Please, provide figures for interventions.

- I am confused about which statistical method is used for age, NRS, and ODI. (line 203-204)

‘’Chi-square with Fisher's exact test was used to compare age, gender, occupation, successful NRS and ODI (%), and the NASS patient satisfaction index after treatment between the two groups. Independent t-tests were performed to identify differences in age, NRS, ODI (%), number of treatment sessions performed

Results

-Please explain the reason for the flow diagram of the study. Why patients did not meet the eligibility criteria. What was the reason for dropouts?

-Please use the CONSORT diagram for clinical trials.

-Was there any adverse effect due to the interventions. Please explain.

Discussion

-The discussion section should also focus on why ‘’ the 4.4-MHz RF diathermy group revealed a higher rate of successful pain relief at 12 weeks after treatment than the US for LBP, but the RF group did not show a significant difference in functional improvement compared to the US group’’

-Similar studies that investigated the effects of RF in LBP should be detailed.

- I don’t think that details of the 4.4-MHz RF diathermy were appropriate for discussion. It may be removed.

Author Response

We would like to thank the reviewers for thoughtful review. We agree with almost your comments and we have revised our manuscript accordingly. Please find our response (in blue) to reviewer’s specific comments (in black) below. The corrected sentences are tracked as word file. We tried to response reviewer’s comment properly and clarifying the text when needed. We hope that the reviewers will find our responses to their comments satisfactory.

[Reviewer 1]

Although this manuscript was written well and has some interesting ideas, there are several significant flaws that need to be edited.

The main problem of the manuscript was the study design. The authors stated out the manuscript was designed as a double-blind, randomized controlled trial aimed to investigate the clinical effectiveness and safety of 4.4-MHz 100 RF diathermy in the treatment of patients with LBP and compare this with the clinical effectiveness of US treatment. However, no control or placebo group was involved. Moreover, I think the ultrasonography group was not a control group, it was an active intervention group. So, the study mainly did not focus on the effect of a 4.4-MHz Radiofrequency Deep Heating Therapy on low back pain. I think the study compares the effect of 4.4-MHz Radiofrequency Deep Heating Therapy and ultrasonography.  According to the study design, the authors can only prove the comparison of Radiofrequency Deep Heating Therapy and ultrasonography. Therefore, the study design should be edited for all sections of the manuscript.

Title

-The title should be revised as ‘’A comparison of the effects of ………………………’’, if a control or placebo group isn’t involved.

[Author’s comment]

Thank you for your thoughtful comment. Your valuable comment will help us to improve the manuscript. As our study investigated the clinical effectiveness of RF diathermy in the treatment of patients with LBP and compare this with the clinical effectiveness of US treatment, we changed the title of manuscript “A Comparison of the Effect of a 4.4-MHz Radiofrequency Deep Heating Therapy and Ultrasound on Low Back Pain: A Randomized, Double-Blind, Multicenter Trial”.

Abstract

- RF was written directly as an abbreviation.

-P values should be added.

[Author’s comment]

Thank you for your thoughtful comment. We changed the sentences according to your comments.

Introduction

- The introduction was well explained. The authors may mention more of the studies investigating the effectiveness of RF in low back pain. The difference and importance of the current study from similar studies can be explained more clearly.

-The purpose of the study should be revised.

[Author’s comment]

Thank you for your comment. In introduction section, we added the study investigating the effectiveness of RF compared to SHT in LBP. And more of study investigating the effectiveness of RF in LBP.

Methods

-I think, the study did not involve any control group, so line 106 should be corrected.

[Author’s comment]

We changed the sentence. “randomized controlled trial was prospectively conducted” to “randomized prospective trial was conducted”.

-Why ages of patients limited from 19 to 70 years? Did the authors have any special causes?

[Author’s comment]

This study was conducted on adults, and elderly people over 70 years of age were excluded because they often had cognitive decline. And as age increases, there is a possibility that there is a difference in recovery for low back pain, so the age was limited in this study. Previous studies have reported an older age associated with a longer time to recovery in low back pain [1].

-Was there a special reason for the ODI score >30% in the inclusion criteria of the patients? Please explain.

[Author’s comment]

The ODI of 21-40% usually indicates moderate disability. It was considered appropriated to confirm the effect of RF in patients with LBP with moderate to severe disability. So we set a value between 21 and 40. 

-Authors stated that patients with 1 to 6 months of low back pain were included in the study. Why did they choose 1 to 6 months? Studies generally include chronic low back pain (more than 3 months) or acute/subacute (less than 3 months). The study did not have a homogenous LBP population.

[Author’s comment]

We agree with your opinion. In this study, we wanted to evaluate the effect of RF diathermy in the patients except for acute low back pain. As you mentioned, subacute and chronic LBP were enrolled, that the study population of this study is not homogenous. We added this to the limitation section.

“Second, we enrolled patients with low back pain between 1 month to 6 months, excluding the acute low back pain. As patients subacute and chronic low back pain were included, the study population of this study is not homogenous.”

-Were the physical activity level or exercise habits of participants available? Were the groups equal physical levels?

[Author’s comment]

We did not measure the activity levels of the study patients. Among the baseline clinical parameters, there was no difference between the two groups in the Biering-Sorensen test and Up and Go test.

-Please, provide figures for interventions.

[Author’s comment]

Yes, we added it as a supplementary figure.

- I am confused about which statistical method is used for age, NRS, and ODI. (line 203-204)

‘’Chi-square with Fisher's exact test was used to compare age, gender, occupation, successful NRS and ODI (%), and the NASS patient satisfaction index after treatment between the two groups. Independent t-tests were performed to identify differences in age, NRS, ODI (%), number of treatment sessions performed

[Author’s comment]

In this study, categorical data were analyzed using Fisher's exact test and chi-square test for comparison between groups, and continuous variables were analyzed using independent t-test. As age distribution, successful rate of NRS and ODI (%) are categorical data, chi-square test was used.

Results

-Please explain the reason for the flow diagram of the study. Why patients did not meet the eligibility criteria. What was the reason for dropouts?

[Author’s comment]

This is cases where a one-experienced clinical research operator before randomization confirmed that it corresponds to study exclusion through detailed history. The reason is a history of stroke (mild impairment), planning to become pregnant, history of back surgery.

-Please use the CONSORT diagram for clinical trials.

[Author’s comment]

Thank you for your comment. We changed the diagram for clinical trials according to CONSORT diagram.

-Was there any adverse effect due to the interventions. Please explain.

[Author’s comment]

There is a case of dropouts due to pregnancy in the RF group. Other than that event, adverse events such as skin problems and burns did not occur in the RF group and US group.

Discussion

-The discussion section should also focus on why ‘’ the 4.4-MHz RF diathermy group revealed a higher rate of successful pain relief at 12 weeks after treatment than the US for LBP, but the RF group did not show a significant difference in functional improvement compared to the US group’’

-Similar studies that investigated the effects of RF in LBP should be detailed.

- I don’t think that details of the 4.4-MHz RF diathermy were appropriate for discussion. It may be removed.

[Author’s comment]

Thank you for your comment. We added in the discussion section about the investigated the effects of RF in LBP. And we reduced and modified the details of the 4.4-MHz RF diathermy.

[Reference]

  1. Henschke, N.; Maher, C.G.; Refshauge, K.M.; Herbert, R.D.; Cumming, R.G.; Bleasel, J.; York, J.; Das, A.; McAuley, J.H. Prognosis in patients with recent onset low back pain in Australian primary care: inception cohort study. Bmj 2008, 337, a171.

Reviewer 2 Report

Dear Author

This study examines the treatment outcomes of RF and US, and the results
show that RF diathermy has significantly better clinical outcomes in
pain relief at 12 weeks post-treatment compared to US. The results of
this study indicate that 4.4 MHz RF diathermy can be used effectively as
a conservative treatment option for LBP patients.

Therefore, this study enhances the evidence for RF as a physical therapy.

On the other hand, the methods, considerations, and limitations need to
be revised.

Major Comments

1. The patients in this study were predominantly women in both groups:
79.0% in RF and 76.8% in US. On the other hand, the discussion did not
focus on this point. This point should be clearly stated in the
limitations of the study, as the factors that cause low back pain may
differ between men and women.
2. L315-「This study evaluated the effectiveness and safety of a 4.4 MHz
RF diathermy in comparison to US for patients with LBP. In this study,
both the RF and US groups showed significantly improved ODI and NRS, as
well as improvements on the Biering Sorensen test and the up and go test
at 4 weeks and 12 weeks after tr eatment. In this study,」

This study used pain and motor function as outcomes and was by no means
a safety evaluation. Therefore, please add the results of the safety
evaluation or revise the text.

Minor Comments

1. Fig2: It is difficult to understand which is pain relief or
functional improvement in FIg2. Please describe it clearly so that the
reader can easily understand. (e.g.; A-1:pain relief 4week, A-2:pain
relief 12week etc...)
2. Table4: As in the other tables, please provide percentages and other
units.
3. In the limitations of the study, it was noted that this study was not
conducted in conjunction with other treatments. This should also be
stated in the Methods.

Author Response

We would like to thank the reviewers for thoughtful review. We agree with almost your comments and we have revised our manuscript accordingly. Please find our response (in blue) to reviewer’s specific comments (in black) below. The corrected sentences are tracked as word file. We tried to response reviewer’s comment properly and clarifying the text when needed. We hope that the reviewers will find our responses to their comments satisfactory.

[Reviewer 2]

This study examines the treatment outcomes of RF and US, and the results
show that RF diathermy has significantly better clinical outcomes in
pain relief at 12 weeks post-treatment compared to US. The results of
this study indicate that 4.4 MHz RF diathermy can be used effectively as
a conservative treatment option for LBP patients.

Therefore, this study enhances the evidence for RF as a physical therapy.

On the other hand, the methods, considerations, and limitations need to
be revised.

Major Comments

1. The patients in this study were predominantly women in both groups:
79.0% in RF and 76.8% in US. On the other hand, the discussion did not
focus on this point. This point should be clearly stated in the
limitations of the study, as the factors that cause low back pain may
differ between men and women.

[Author’s comment]

Thank you for your thoughtful comment. Your valuable comment will help us to improve the manuscript. We added in the limitation “Third, the physical and physiology features that causes LBP may differ between men and women. In this study were predominantly women in both group, from this it is difficult to generalize. A limitation in this study is that it was predominantly female in both groups”.

  1. L315-「This study evaluated the effectiveness and safety of a 4.4 MHz
    RF diathermy in comparison to US for patients with LBP. In this study,
    both the RF and US groups showed significantly improved ODI and NRS, as
    well as improvements on the Biering Sorensen test and the up and go test
    at 4 weeks and 12 weeks after treatment. In this study,」

    This study used pain and motor function as outcomes and was by no means
    a safety evaluation. Therefore, please add the results of the safety
    evaluation or revise the text.

[Author’s comment]

Thank you for your thoughtful comment. We revised the sentence “This study evaluated the effectiveness of a 4.4-MHz RF diathermy in comparison to US for patients with LBP”. During study, there is a case of dropouts due to pregnancy in the RF group. Other than that event, adverse events such as skin problems and burns did not occur in the RF group.

Minor Comments

1. Fig2: It is difficult to understand which is pain relief or
functional improvement in FIg2. Please describe it clearly so that the
reader can easily understand. (e.g.; A-1:pain relief 4week, A-2:pain
relief 12week etc...)

[Author’s comment]

We changed Fig 2. And we added the comments “A. Successful rate of pain relief at 4 weeks, B. Successful rate of pain relief at 12 weeks, C. Successful rate of functional improvement at 4 weeks, D. Successful rate of functional improvement at 4 weeks”.

  1. Table4: As in the other tables, please provide percentages and other
    units.

[Author’s comment]

We revised the Table 4.

  1. In the limitations of the study, it was noted that this study was not
    conducted in conjunction with other treatments. This should also be
    stated in the Methods.

[Author’s comment]

Thank you for your comments. We added it in the Methods “In this study, RF or US treatment was applied to each group alone, and other treatments including exercise therapy were not administered together”.

Reviewer 3 Report

Article

Effect of a 4.4-MHz Radiofrequency Deep Heating Therapy on Low Back Pain: A Randomized, Double-Blind, Multicenter Trial

GENERAL COMMENTS

Thank you for allowing me to review this manuscript. The manuscript adhere the JCM Data Policy. The aim of this study was to investigate the effectiveness of 4.4-MHz 37 RF diathermy compared to ultrasound (US) in patients with LBP. A double-blind, randomized controlled trial was prospectively conducted with 118 patients in three hospitals of a musculoskeletal clinic.

This is an interesting research topic with potential utilization across disciplines and relevant to the journal. In my opinion, the paper would need minor changes. Revisions will be necessary. Improve the organization of your paper using the following guidelines.

INTRODUCTION

- Abstract: The title of the study and the objective of the study is not matching (include: compared to ultrasound (US)) please revise it.

- What is the current state of evidence on the therapeutic approach for the condition studied in the study? What's new in the scientific literature with this manuscript? Include in introduction. How well is the paper integrated with current research.

-The introduction needs to better indicate the existing evidenceon therapeutic approaches for the condition studied. The use of systematic reviews is recommended.

-The manuscript  include a hypothesis: “Previous studies reported that the  therapeutic effect of a 4.4-MHz RF on muscle injury in a rat model might be due to the anti inflammatory action of RF . For this reason, we set the RF frequency to 4.4 MHz. We  hypothesized that a 4.4-MHz RF diathermy would be effective to decrease pain and improve physical function in patients with LBP. HOW? WHY?  Include. Explain the hypothesis.

METHODS

- Please follow the CONSORT – Guidelines to present the article.

- Who has diagnosed the condition and on what basis or criteria he has been diagnosed?

-Include the ICD classification of the disease.

-Mention the treatment adherence rate, adverse effects, and a number of drops out during the study.

RESULTS

- The results should be presented with 95%CI (upper limit – lower limit) for all the variables.

DISCUSSION

-Include the strengths of the study.

-Thus, the reader can decide whether the results have relevance and applicability in their daily practice. Question? ARE THE RESULTS APPLICABLE IN PRACTICE? Add. Include the implications/applications of the study for the field movement sciences. The final paragraph should leave the reader with your final message within the framework of the hypotheses posed in the Introduction.

- Include the clinical significance of this study over clinicians, patients, and researchers after the study hypothesis.

Author Response

We would like to thank the reviewers for thoughtful review. We agree with almost your comments and we have revised our manuscript accordingly. Please find our response (in blue) to reviewer’s specific comments (in black) below. The corrected sentences are tracked as word file. We tried to response reviewer’s comment properly and clarifying the text when needed. We hope that the reviewers will find our responses to their comments satisfactory.

[Reviewer 3]

GENERAL COMMENTS

Thank you for allowing me to review this manuscript. The manuscript adhere the JCM Data Policy. The aim of this study was to investigate the effectiveness of 4.4-MHz 37 RF diathermy compared to ultrasound (US) in patients with LBP. A double-blind, randomized controlled trial was prospectively conducted with 118 patients in three hospitals of a musculoskeletal clinic.

This is an interesting research topic with potential utilization across disciplines and relevant to the journal. In my opinion, the paper would need minor changes. Revisions will be necessary. Improve the organization of your paper using the following guidelines.

INTRODUCTION

- Abstract: The title of the study and the objective of the study is not matching (include: compared to ultrasound (US)) please revise it.

[Author’s comment]

Thank you for your thoughtful comment. Your valuable comment will help us to improve the manuscript. As our study investigated the clinical effectiveness of RF diathermy in the treatment of patients with LBP and compare this with the clinical effectiveness of US treatment, we changed the title of manuscript “A Comparison of the Effect of a 4.4-MHz Radiofrequency Deep Heating Therapy and Ultrasound on Low Back Pain: A Randomized, Double-Blind, Multicenter Trial”.

- What is the current state of evidence on the therapeutic approach for the condition studied in the study? What's new in the scientific literature with this manuscript? Include in introduction. How well is the paper integrated with current research.

[Author’s comment]

Thank you for your comments. We have added a description of the treatment for low back pain and described the need for research in introduction section.

-The introduction needs to better indicate the existing evidence on therapeutic approaches for the condition studied. The use of systematic reviews is recommended.

[Author’s comment]

Thank you for your comments. We added in the introduction section using systemic review.

-The manuscript  include a hypothesis: “Previous studies reported that the  therapeutic effect of a 4.4-MHz RF on muscle injury in a rat model might be due to the anti inflammatory action of RF . For this reason, we set the RF frequency to 4.4 MHz. We  hypothesized that a 4.4-MHz RF diathermy would be effective to decrease pain and improve physical function in patients with LBP. HOW? WHY?  Include. Explain the hypothesis.

 [Author’s comment]

Thank you for your thoughtful comment. We added in the introduction section “A 4.4-MHz RF elevated the muscle layer temperature without inducing cellular damage. Furthermore, a 4.4-MHz RF showed therapeutic effect on muscle contusion by reducing muscle swelling in a rat model. This has the advantage of a tissue penetration without tissue damage and an anti-inflammation action in the target tissue.”.

METHODS

Please follow the CONSORT – Guidelines to present the article.

[Author’s comment]

Thank you for your comment. We changed the diagram for clinical trials according to CONSORT diagram (figure 1).

Who has diagnosed the condition and on what basis or criteria he has been diagnosed?

[Author’s comment]

Physical and Rehabilitation medicine doctors of 3 hospitals that have operated musculoskeletal clinics for more than 10 years made the diagnosis. Simple lumbar X-ray, Laboratory test (including ESR, CRP) were performed, and diagnosis was made through the patient's physical examination and medical history.

-Include the ICD classification of the disease.

[Author’s comment]

We added ICD classification of the Low back pain in Introduction section.

-Mention the treatment adherence rate, adverse effects, and a number of drops out during the study.

 [Author’s comment]

We added the number of drop out during the study, and the number of incompletion of treatment.  There is a case of dropouts due to pregnancy in the RF group. Other than that event, adverse events such as skin problems and burns did not occur in the RF group and US group.

RESULTS

The results should be presented with 95%CI (upper limit – lower limit) for all the variables.

 [Author’s comment]

Thank you for your comment. We added 95% CI for the primary and secondary outcomes.

DISCUSSION

-Include the strengths of the study.

 [Author’s comment]

Thank you for your comment. We added in discussion section “The strength of this study was that the relatively large sample size, double blind, randomized study design provided a better understanding of the clinical efficacy of RF diathermy in LBP. And in our study, RF diathermy and US were applied as monotherapy rather than in combination with other treatments (i.e., exercise and/or manual therapy). Through this study design, we tried to avoid the conflicting effects of different treatments in order to weigh the effectiveness of each single treatment.”.

-Thus, the reader can decide whether the results have relevance and applicability in their daily practice. Question? ARE THE RESULTS APPLICABLE IN PRACTICE? Add. Include the implications/applications of the study for the field movement sciences. The final paragraph should leave the reader with your final message within the framework of the hypotheses posed in the Introduction.

 [Author’s comment]

Thank you for your comment. We added it in the last paragraph of the discussion section.

- Include the clinical significance of this study over clinicians, patients, and researchers after the study hypothesis.

 [Author’s comment]

Thank you for your comment. It is described in the first paragraph of the discussion section and conclusion.

Reviewer 4 Report

This study was to assess the efficacy of a 4,4 MHz radiofrequency deep heating therapy on low back pain by comparing US. The authors adopted a high level of evidence to assess the efficacy of radiofrequency. The authors concluded that the radiofrequency could be used to treat LBP. However, some considerations should be draw in this manuscript. 

1.     Line 58 Please cited a reference.

2.     Table 1 should be deleted, because it did not provide too much information and the information should be described in the text.

3.     NRS should be considered as primary outcome, because the major cause of function is pain. The purpose of these therapies is to reduce LBP. 

4.     Lines 191-192, do you have any reference to support this definition?

5.     In Table 2, why the authors classified age into three subgroups?

6.     In Table 2, the authors should note which statistical tests were adopted.

7.     The authors should combine Table 2 and Table 3 into in one Table.

8.     In section 3.2, the authors should cite the Table 4 in the descriptions when a significant difference or non-significant difference was observed. 

9.     In Table 4, the authors should note more clearly. For example, a is paired t-test, but I cannot know what pair do you consider. b indicates independent t-test but what the two variables were compared. C is repeated ANOVA, but what did you compare?

10.  Line 308, the authors said that no serious side effects were observed. Did minor side effects were observed in either group?

11.  The P-value in Line 305 was not the same with that in Table 5.

12.  In the study limitations, the 1st seems not to be a limitation. It helps us to clear the efficacy between two therapies. The 2nd limitation, I can tell any study limitation in these descriptions. Additionally, taking medicine during treatment should be a limitation, but the authors did not mention it.

Author Response

We would like to thank the reviewers for thoughtful review. We agree with almost your comments and we have revised our manuscript accordingly. Please find our response (in blue) to reviewer’s specific comments (in black) below. The corrected sentences are tracked as word file. We tried to response reviewer’s comment properly and clarifying the text when needed. We hope that the reviewers will find our responses to their comments satisfactory.

[Reviewer 4]

This study was to assess the efficacy of a 4,4 MHz radiofrequency deep heating therapy on low back pain by comparing US. The authors adopted a high level of evidence to assess the efficacy of radiofrequency. The authors concluded that the radiofrequency could be used to treat LBP. However, some considerations should be draw in this manuscript. 

  1. Line 58 Please cited a reference.

[Author’s comment]

Thank you for your thoughtful comment. We added the reference.

  1. Table 1 should be deleted, because it did not provide too much information and the information should be described in the text.

[Author’s comment]

Thank you for your comment. We deleted the table 1.

  1. NRS should be considered as primary outcome, because the major cause of function is pain. The purpose of these therapies is to reduce LBP. 

[Author’s comment]

   Thank you for your valuable comment. As you mentioned, pain is an important indicator of treatment effectiveness. From the perspective of PMR doctor, this study focused on low back pain and the resulting impairments in activity and participation. In this regard, we set ODI as the primary outcome, and NRS and several functional measures as secondary outcomes.

  1. Lines 191-192, do you have any reference to support this definition?

[Author’s comment]

Thank you for your comment. In previous study, it was used to set the outcome of the effect in spine disease. The reference is attached “Significant pain relief was described as 50% or more reduction in the NRS from baseline, whereas significant improvement in function was described as at least a 40% reduction in the ODI” [1].

  1. In Table 2, why the authors classified age into three subgroups?

[Author’s comment]

Thank you for your comment. As age increases, there is a possibility that there is a difference in recovery for low back pain, so the age was corrected. Previous studies have also reported an older age associated with a longer time to recovery in low back pain [2]. In general, adult means up to age 39, middle age adult 40 - 59 yrs, and seninor adult age 60 or older, so we divided them into three groups.

  1. In Table 2, the authors should note which statistical tests were adopted.

[Author’s comment]

Thank you for your comment. We added in the table 2 “Chi-square with Fisher's exact test was used to compare gender, age distribution, occupation. Independent t-tests were performed to identify differences in age, height, weight.”

  1. The authors should combine Table 2 and Table 3 into in one Table.

[Author’s comment]

Thank you for your comment. I would like to present baseline data separately from demographic and clinical parameters. Table 2 presents the demographic characteristics of the entire patient group, and Table 3 presents clinical parameters between groups. If merging the tables will help readers, I will.

  1. In section 3.2, the authors should cite the Table 4 in the descriptions when a significant difference or non-significant difference was observed. 

[Author’s comment]

Thank you for your comment. We added comment in section 3.2 “The changes in clinical and functional parameters in both groups are shown in Table 3”.

  1. In Table 4, the authors should note more clearly. For example, a is paired t-test, but I cannot know what pair do you consider. b indicates independent t-test but what the two variables were compared. C is repeated ANOVA, but what did you compare?

  [Author’s comment]

Thank you for your comment. We need to explain clearly, So I added “ aPaired t-test (within groups, baseline vs. at 4 weeks, baseline vs. at 12 weeks), b Independent t-test (between groups, RF vs. US), c Time effect - Repeated measures of ANOVA between groups analysis was applied: Baseline, 4 and 12 weeks after treatment between groups”.

  1. Line 308, the authors said that no serious side effects were observed. Did minor side effects were observed in either group?

[Author’s comment]

There is a case of pregnancy in the RF group. Other than that event, adverse events such as skin problems and burns did not occur in the RF group and US group.

  1. The P-value in Line 305 was not the same with that in Table 5.

[Author’s comment]

It was corrected because there was an error in the table.

  1. In the study limitations, the 1stseems not to be a limitation. It helps us to clear the efficacy between two therapies. The 2nd limitation, I can tell any study limitation in these descriptions. Additionally, taking medicine during treatment should be a limitation, but the authors did not mention it.

[Author’s comment]

Thank you for your important comment. I agree with your opinion. We added in limitation section “Lastly, it is a limitation of the study that this study did not prohibit of oral pain medication. However, we showed administered pain medication as a secondary outcome”.

[References]

  1. Manchikanti, L.; Singh, V.; Cash, K.A.; Pampati, V.; Datta, S. A comparative effectiveness evaluation of percutaneous adhesiolysis and epidural steroid injections in managing lumbar post surgery syndrome: a randomized, equivalence controlled trial. Pain Physician 2009, 12, E355-368.
  2. Henschke, N.; Maher, C.G.; Refshauge, K.M.; Herbert, R.D.; Cumming, R.G.; Bleasel, J.; York, J.; Das, A.; McAuley, J.H. Prognosis in patients with recent onset low back pain in Australian primary care: inception cohort study. Bmj 2008, 337, a171.

Round 2

Reviewer 1 Report

The authors' responses and corrections were quite adequate. It is now a well-designed and well-written manuscript. In my opinion, this manuscript may be accepted.

Author Response

We would like to thank for your thoughtful review. 

Reviewer 4 Report

The authors have solved my concerns regarding this research. The revised version could be accepted.

Author Response

(The authors gave the same response as above.)
